# Bone Morphogenetic Protein 2 Promotes Bone Formation in Bone Defects in Which Bone Remodeling Is Suppressed by Long-Term and High-Dose Zoledronic Acid

**DOI:** 10.3390/bioengineering10010086

**Published:** 2023-01-09

**Authors:** Young Jae Moon, Seongyup Jeong, Kwang-Bok Lee

**Affiliations:** 1 Department of Biochemistry and Orthopaedic Surgery, Jeonbuk National University Medical School and Hospital, Jeonju 54896, Jeonbuk, Republic of Korea; 2 Research Institute of Clinical Medicine of Jeonbuk National University-Biomedical Research Institute of Jeonbuk National University Hospital, Jeonju 54896, Jeonbuk, Republic of Korea; 3 Department of Orthopaedic Surgery, Jeonbuk National University Medical School and Hospital, Jeonju 54896, Jeonbuk, Republic of Korea

**Keywords:** bisphosphate, zoledronic acid, bone morphogenetic protein, bone defect, bone remodeling, atypical fracture

## Abstract

The use of long-term and high-dose bisphosphate is associated with severely suppressed bone turnover and the delayed union of fractures. However, therapeutic methods to overcome the negative effects of bisphosphonate use are lacking. Bone morphogenetic proteins (BMPs) are powerful osteoinductive proteins. The development of the delivery system using BMP has been verified to have an excellent effect on fracture healing and the enhancement of osteointegration. We hypothesized that BMPs had similar effects as autografts in patients with decreased bone healing potential due to long-term bisphosphonate treatment. Forty rats were divided into the following four groups depending upon the materials implanted into the femoral defect after ten weeks of bisphosphonate (zoledronic acid) injections: Group I: absorbable collagen sponge (control); group II: demineralized freeze-dried bone graft; group III: autogenous bone graft; and group IV: rhBMP-2 with an absorbable collagen sponge. Radiographic union, micro-computed tomography (CT) analysis, manual palpation, and histologic analysis were evaluated. The radiographic union rate, manual union rate, and micro-CT bone volume in groups III and IV were significantly higher than those in groups I and II. Groups III and IV showed similar results to each other. Although the amount of immature bone in the BMP-treated group was large, the effect was similar to that of autografts in the bone defect model in which bone turnover was severely reduced by bisphosphonate treatment. BMP might be a good substitute for autografts in patients with decreased bone healing potential due to long-term bisphosphonate treatment.

## 1. Introduction

Bisphosphonates are widely used to treat osteoporosis because they are believed to prevent bone loss by inhibiting osteoclast activity [1]. These drugs increase bone strength and reduce the risk of fracture in patients with osteoporosis, Paget’s disease, and metastatic bone disease [2,3,4]. However, because osteoclasts have a major role in the fracture repair and remodeling of bone graft sites, long-term and high-dose bisphosphonate use impaired callus remodeling and the healing of bone defect sites in animal models [5,6,7]. Clinically, the long-term use of bisphosphonates in osteoporotic patients or high doses of bisphosphonates used for the treatment of osteogenesis imperfecta and metastatic bone tumors leads to atypical femoral fractures [8,9,10]. In addition, case series have shown that the long-term use of bisphosphonates increased the occurrence of nonunions after intramedullary nailing treatment for atypical femur fractures [11,12]. However, there are still limited treatment choices after an atypical fracture due to long-term and high-dose bisphosphonate treatment. 

Bone morphogenetic proteins (BMPs) are a subgroup of the transforming growth factor-β (TGF-β) superfamily [13,14,15]. Many studies have reported that BMPs can accelerate bone regeneration and fracture healing [14,16]. Furthermore, BMPs are as effective as autogenous bone grafts for fracture healing [17]. The potential benefits of using BMPs include decreased donor-site morbidity associated with harvesting the autograft and reduced costs associated with treatment for delayed unions and nonunions [18]. Also, through recent studies, delivery systems (polyelectrolyte multilayer coating method on implants, scaffold with high affinity to BMP) that can minimize the side effects of BMP are being developed, increasing the effective use of BMP in the future [19,20,21].

We hypothesized that BMPs had a similar effect as autografts on the union rate of fractures in patients with decreased bone healing potential due to long-term and high-dose bisphosphonate treatment. Therefore, the objective of this study was to compare BMPs with demineralized freeze-dried bone (DFDB) and autografts in a rat femoral defect model with long-term and high-dose bisphosphonate treatment.

## 2. Materials and Methods

### 2.1. Study Design

This study was approved by the Institutional Animal Care and Use Committee (IACUC) of the Jeonbuk National University Laboratory Animal Center, Jeonbuk National University, Republic of Korea (CBU 2014-00077). All animals were cared for in accordance with the regulations of the IACUC under the supervision of licensed veterinarians. The study was conducted according to the ARRIVE guidelines for the reporting of animal experiments. The surgical procedures and assessment of the unions were performed according to the methods of Liao et al. [13]. 

Forty rats were divided into the following four groups, and ten rats were assigned to each group (Figure 1a):

Group I: absorbable collagen sponge (ACS) (Gelfoam; Mascia Brunelli Spa, Milano, Italy) alone group; group II: DFDB graft group-cancellous bone chips (Community Tissue Services, Dayton, OH, USA); group III: autogenous bone graft group (iliac cortico-cancellous bone); group IV: rhBMP-2 implant group (Daewoong CG Bio Pharmaceutical Co., Ltd., Seoul, Republic of Korea) with ACS. We used collagen sponges as BMP delivery scaffolds because they are widely used in clinical and are FDA-approved [22]. To confirm the effect of BMP alone, the group in which the collagen sponge was inserted into the bone defect was used as a control group (group I). 

The dose of rhBMP used for the bone defect was 10 µg. This is because 10 µg was the most effective inducing bone formation dose in the rat femoral defect model [23,24]. Pathogen-free 12-week-old female Lewis rats (230–290 g) were purchased from Orient Bio (Seoul, Republic of Korea). They were housed in a laminar flow cabinet with a 12 h light/dark cycle and maintained on standard laboratory chow ad libitum. Each rat was given an intraperitoneal injection of 0.1 mg/kg zoledronic acid (Zometa) once a week for 10 weeks. After that, femoral bone defect surgery was performed on all rats. All rats underwent radiographic studies every 4 weeks after the surgical procedure. Sixteen weeks after the operation, the bone volume in the defect site of the femur was assessed by micro-computed tomography. All animals were euthanized 16 weeks after surgery, and their right femurs were harvested. Union rate determination, manual palpation, and histologic studies were performed for the defect site of the femur (Figure 1b). 

### 2.2. Surgical Procedures

All rats were anesthetized using Zoletil® (Virbac, Carros, France) intraperitoneally (7 mg/kg). The right hind limb was shaved and disinfected with alcohol and povidone-iodine. A longitudinal 3-cm skin incision was made in the lateral aspect of the femur, and the entire length of the right femur was exposed. The periosteum was stripped from the shaft. A 5 mm mid-diaphyseal full-thickness defect was created using a bone saw. A polyethylene plate (23 mm × 4 mm × 4 mm) was secured with 4 0.99 mm Kirschner wires and 2 0.53 mm steel cerclage wires on the lateral side of the femur, as previously described [13]. Each testing material was inserted into the defect in groups II, III, and IV. ACS only was inserted into group I. Iliac cortico-cancellous bone was used for the autograft group. We performed an oblique 2-cm skin incision over the posterior superior iliac spine (PSIS) to retract the soft tissue and expose the PSIS. After cortical osteotomy, cortico-cancellous bone grafts were taken using a small-size curette. Autobone and DFDB were grafted in 0.5 cc amounts. For accurate grafting, we measured 0.5 cc using a 2-cc syringe. rhBMP was implanted at 10 µg/0.5 cc.

### 2.3. Radiographic Evaluation

All animals were anesthetized, and radiographs were taken every 4 weeks. The rats were imaged with a high-resolution digital mammographic imager (Mammomat Novation, Siemens AG Medical Solutions, Erlangen, Germany). All images were obtained with exposure settings of 34 kVp and 110 mA at a magnification of 1.5×. Radiographic union was assessed by bony bridge formation of the cortex. A complete union was defined as the bony bridge formation of both cortexes in radiography, while a partial union was defined as the bony bridge formation of one cortex. Nonunion was defined as no bony bridge formation.

### 2.4. Micro-CT Analysis

Sixteen weeks after surgery, all rats underwent micro-CT scanning (NFR Polaris-G90, NanoFocusRay Co., Ltd., Jeonju, Republic of Korea) to evaluate bone volume and healing of the defect site. The scanner was set at an X-ray voltage of 70 kVp with an X-ray current of 100 µA. The scans were completed over 360° of rotation of the X-ray tube. All micro-CT image data were acquired using live, free-breathing rats anesthetized by intraperitoneal Zoletil® injections. We measured the bone volumes around the defect site using a micro-CT scanner (NFR Polaris-G90). A cylindrically shaped region of interest (ROI) was established with the middle of the bone defect site as the center, which had a diameter of 14.25 mm and a height of 6.65 m. The bone volume (BV), tissue volume (TV), and bone volume (BV/TV) percentages were measured in the ROI.

### 2.5. Manual Palpation and Manipulation

Manual palpation is a sensitive and specific method of assessing bone fusion [13,25,26,27]. Manual palpation with varus/valgus and anterior/posterior angulated force was performed for all harvested specimens with a particular focus on the defect region. All specimens were manipulated with a force high enough to evaluate the gross motion. When no motion was present, it was considered a complete union, as previously described [13].

### 2.6. Histology Analysis

We performed bony biopsies of the resected femurs to evaluate the bone healing potential. The resected femurs were fixed in 10% neutral buffered formalin and decalcified in 10% ethylenediaminetetraacetic acid (EDTA) for 10 days or in a rapid decalcifying solution (Calci-Clear Rapid, National Diagnostics, Atlanta, GA, USA) for 12–24 h. To evaluate the histologic changes, paraffin-embedded tissue sections were stained with hematoxylin and eosin (Sigma-Aldrich, St. Louis, MO, USA). The area of newly formed bone was measured using the iSolution DT36 software (Carl Zeiss, Oberkochen, Germany).

### 2.7. Statistical Analysis

Statistical analysis was carried out with Statistical Package for Social Sciences software (SPSS Inc., Seoul, Republic of Korea). The union rates detected by radiological evaluation (union rate), the manual test, and the bone volume in micro-computed tomography were compared using one-way analysis of variance (ANOVA) with a posthoc test (Scheffe test). *p*-values of less than 0.05 were considered statistically significant.

## 3. Results

All rats tolerated the surgical procedures. There were no serious complications after the surgeries.

### 3.1. The Effect of Long-Term Treatment with Bisphosphonate

Long-term and high-dose bisphosphonate injections for osteoporosis treatment are associated with complications such as osteonecrosis characterized by empty lacunae in the deeper layer of bone and chronic inflammation in the bone marrow [28]. To confirm whether long-term and high-dose bisphosphonate worked in this model, we histologically observed the distal femur of rats treated with long-term and high-dose zoledronate. As expected, empty lacunae were observed in more than 50% of the osteocyte lacunae in the distal femur, and the infiltration of chronic inflammatory cells was observed in the bone marrow cavity (Figure 2). Furthermore, metaphyseal bands, which are called zebra lines, were observed in the proximal tibia (Figure 3; yellow arrows). Zebra lines were observed near the ends of the long bones during cyclic bisphosphonate therapy for osteogenesis imperfect [29].

### 3.2. Radiographic Evaluation 

The results of the radiographic evaluation are summarized in Table 1. Sixteen weeks after the surgical procedure, complete radiographic unions were achieved in six rats in group III, six rats in group IV, two rats in group II, and none in group I. Partial unions were achieved in one, two, three, and two rats of groups I, II, III, and IV, respectively (Figure 3). The partial and complete union rates of groups III and IV were significantly higher than those of groups I and II. There were no significant differences between groups III and IV and between groups I and II. 

### 3.3. Manual Palpation and Manipulation 

The results of the manual palpation for union (Table 2) were similar to the radiographic results. Complete unions were achieved in six rats in group III, six rats in group IV, two rats in group II, and none in group I. In the manual test, the union rates of the rats in groups III and IV were significantly (*p* < 0.05) higher than those of groups I and II. 

### 3.4. Micro-CT Analysis—Bone Volume 

The three-dimensional (3D) reconstruction images are shown in Figure 4. The results of the micro-CT analysis for bone volume are shown in Table 3. The mean bone volumes in groups I, II, III, and IV were 42.8 ± 19.8 mm³, 72.93 ± 15.6 mm³, 168.78 ± 20.1 mm³, and 193.94 ± 61.5 mm³, respectively. The mean bone volumes of groups III and IV were significantly (*p* < 0.05) larger than those of groups I and II. However, there was no significant (*p* > 0.05) difference in bone volume between groups III and IV. The mean bone volume per tissue volume (BV/TV) percentage in the I, II, III, and IV groups was 15.79 ± 5.6%, 36.44 ± 2.5%, 64.3 ± 1.4%, and 56.59 ± 9.8%, respectively. The mean BV/TV of groups III and IV was significantly higher than that of groups I and II. In contrast to the bone volume pattern, the mean BV/TV value in group III was increased compared to that in group IV, but the increase was not significant. These results suggested that the autobone grafts had a small amount of callus, but the remodeling potential was faster than that of BMP. In addition, these results suggest that bone formation has occurred beyond the bone defect site where the femur was initially located (heterotopic ossification, Figure 4d) [21].

### 3.5. Histological Analysis 

In the histological analysis of the resected femurs, the bone defect sites in the absorbable collagen sponge alone group were filled with fibrous tissue without bone formation. In the DFDB graft group, the defect site was filled with fragmented lamellar bone. Lamellar bone in the defect sites of the bone autografts was more abundant and thicker, with relatively less fibrous tissue compared to that of the DFDB graft group. However, the junction between the cortical bone and the newly formed bone in the autografts was partially discontinuous. In contrast, the defect site in the BMP group was filled with newly formed bone, and the junction between the cortical bone and new bone was continuous (Figure 5a). Furthermore, similar to the micro-CT analysis, the area of new bone performed on tissue sections was measured the widest in the BMP group (Figure 5b).

## 4. Discussion

Our findings demonstrated the potential of rhBMP for bone formation, even in bone defects in which bone remodeling was severely reduced by long-term and high-dose zoledronic acid treatment. Although the BMP-induced bone formation seemed immature compared to that of the autogenous bone grafts, the manual tests and histological findings showed that BMP had a similar effect on the bony union as autogenous bone grafts in severely suppressed bone turnover conditions. Based on these results, BMP might be used as an effective substitute for autografts in patients with long-term and high bisphosphonate treatment. 

The novel finding of this study was the effect of BMP on bone defects in rats with severely suppressed bone turnover. Recent studies on the association of bisphosphonate and BMP reported a positive effect of bisphosphonate in the inhibition of osteoclast activity, compared between a group treated with BMP alone and a group treated with bisphosphonates and BMP simultaneously after the fracture [25,30,31]. These studies focused on the temporary suppression of osteoclast activity after a fracture. However, clinically, long-term and high-dose bisphosphonates use inhibited bone turnover, accumulated micro-fractures, reduced energy absorption, increased the fragility of osteoporotic bones, and inhibited bone remodeling through osteocyte death, eventually leading to atypical fractures [6,9,32]. It also promoted nonunion after the treatment of these fractures [11,12]. The present study focused on these side effects of bisphosphonates and confirmed the role of BMP. 

Although this study did not experimentally prove the mechanism of bone formation regulation by BMP in an environment where osteoclast activity was excessively inhibited, it can be speculated through other studies. Because osteoblasts and osteoclasts are coupled, long-term high-dose bisphosphonate inhibits the function of osteoblasts as well as osteoclasts [6]. However, bone regeneration and bone formation in bone defects or fractures occur from periosteal stem cells and bone marrow stem cells around the injured site [33]. Since the cellular viability of these skeletal stem cells was not affected by bisphosphonate [34], it is thought that BMP effectively promoted the osteogenesis of skeletal stem cells recruited to the damaged area. In addition, excessive bone formation was observed in the BMP-treated group, which is thought to be due to the suppression of the function of osteoblasts and osteoclasts by bisphosphonates, thereby reducing the remodeling of new bone. 

In the present study, in the group soaked with BMP on an absorbable collagen sponge, bone volume increased, but bone volume per tissue volume decreased compared to the autogenous bone graft group, and bone formation was observed in the broader area than the original bone defect area on 3D CT. These observations are considered equivalent to previously published side effects of collagen sponges delivering BMP. Previous reports suggested that BMP delivery using collagen sponges could induce ectopic bone formation and irregular bone tissue in bone fusion sites and bone defect animal models [20,21,35]. However, recently developed biomaterials have been shown to reduce heterotopic ossification that occurs when using collagen sponges in animal models [20,21]. In the future, a BMP delivery system using a new material will be applied to confirm the effect of BMP in a long-term, high-dose bisphosphonate bone defect model. 

BMP is produced from osteoprogenitor cells, osteoblasts, chondrocytes, and platelets during the bone healing process [14,36]. The role of endogenous BMP-2 produced by fractures has been confirmed to be essential for fracture healing, as evidenced by delayed fracture healing in mice genetically depleted of BMP-2 [37]. Thus, we may ask whether endogenous BMP production after bone defect surgery is affected by long-term high-dose bisphosphonates. This study did not include the collagen sponges-only group that was not treated with bisphosphonates, the role of endogenous BMPs could not be accurately identified. However, compared with other studies that analyzed bone formation by inserting only a collagen sponge in a bone defect model [38,39], a similar level of bone formation was observed in the collagen sponge-implanted bone defect treated with a bisphosphonate, suggesting that endogenous BMP was not significantly affected by bisphosphonates. Still, this experiment could not confirm the evaluation of the positive feedback reaction [40] (response of endogenous BMP against exogenous BMP) that occurs after treating exogenous BMP.

For surgeries that need bone grafts, such as cases of nonunion or osteotomy, most surgeons may perform autogenous bone grafts. However, the osteogenic effect and the amount of autogenous bone used are lower in older patients than in younger patients [41]. Moreover, in older patients with long-term bisphosphonate treatment, the union rate of osteotomies or nonunions might be low even if an autogenous bone graft is undertaken [42]. Therefore, alternatives to autogenous bone are needed. Our results suggested that BMP might be a suitable substitute for autogenous bone in patients with poor bone quality. Supporting our results, the efficacy of BMP as an enhancer of bone repair has already been confirmed in several studies of segmental bone defects or fractures [43,44]. Multiple clinical trials have shown that BMP and autogenous bone grafts had similar fracture healing rates, with reduced blood loss and shortened surgery time, without donor site morbidities [16,45]. 

This study had several limitations. First, it is unclear whether bone healing was actually reduced in the femur in the femoral defect model in this study. To resolve this limitation, we performed intraperitoneal injections of 0.1 mg/kg zoledronic acid once a week for ten weeks, according to a previous study [46]. Furthermore, through bone biopsy and radiographic analysis of the femur, we confirmed empty lacunae and zebra lines. Therefore, we would indirectly ensure that the femur bone healing ability and bone turnover rate were reduced. Second, the rats used in this study were 12 weeks old. Since most humans treated with long-term bisphosphonates are elderly patients with osteoporosis, the rats used in this study may not fully represent the elderly patients targeted in this study. Third, we did not include a group of rats that did not receive zoledronic acid and groups with different concentrations of BMP. Because it had already been reported that BMP facilitated bone healing and 10 ug of BMP was the optimal dose in the femoral bone defect model, we did not include a zoledronic acid non-treated group and used 10 ug of BMP in the experiments [23,24]. Fourth, we did not perform biomechanical testing that could quantify the degree of bony union. Instead, we evaluated the bony union by the manual palpation method [13,25,26,27]. 

## 5. Conclusions

We showed that BMP and autografts have similar effects on bone formation in animal models with severe bone turnover defects via long-term and high-dose bisphosphate injection. Although the bone defect site where the BMP with collagen sponge was grafted became excessive bone formation and irregular bone shape was observed, it is thought that it will be improved by applying the advanced BMP delivery system. These results suggest that BMP is an option for treating nonunion following a fracture in patients with severely inhibited bone turnover, such as long-term treatment of bisphosphonates. In addition, these results suggested that BMP is effective in atypical fractures, a complication that may occur in patients who have used long-time bisphosphonate treatment.

## Figures and Tables

**Figure 1 bioengineering-10-00086-f001:**
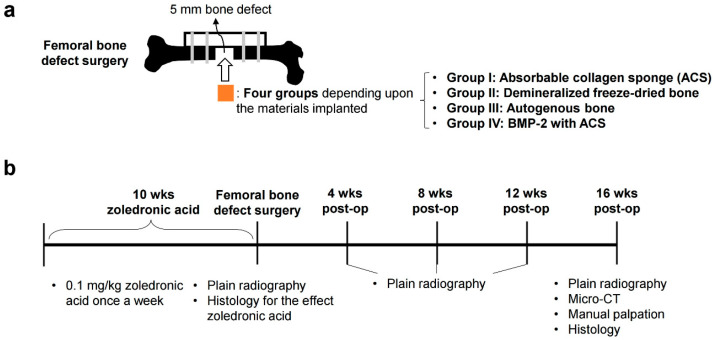
A schematic diagram of study design. (**a**) The schematic figure of femoral bone defect surgery. Each animal was performed with a 5 mm femoral bone defect, stabilized by a polyethylene plate with K-wire. The defects were implanted with an absorbable collagen sponge (group I), with demineralized freeze-dried bone (group II), with autogenous bone (group III), or with BMP-2 with an absorbable collagen sponge (group IV). (**b**) The study’s timeline indicates the time points for injection of zoledronic acid, femoral bone defect surgery, radiology, histology, micro-CT, and manual palpation.

**Figure 2 bioengineering-10-00086-f002:**
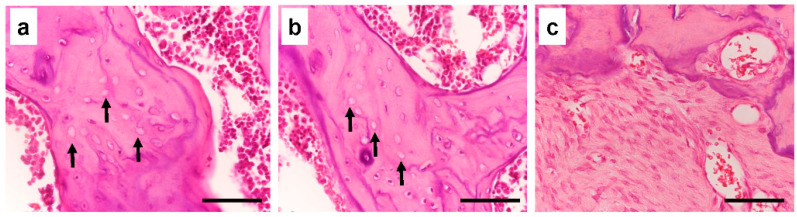
Histological analysis of distal femur. Cancellous bone shows focal osteonecrosis represented by empty lacunae ((**a**,**b**): arrows) and fibrotic proliferation in the bone marrow (**c**). Black bars = 50 μm.

**Figure 3 bioengineering-10-00086-f003:**
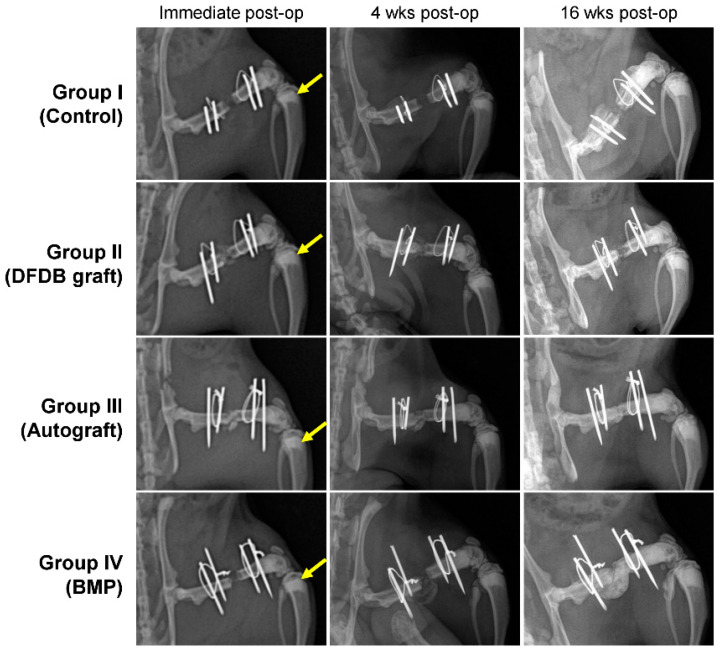
Radiographic findings at postoperative four and 16 weeks in each group. Groups I (Control) and II (DFDB graft) showed nonunion, while groups III (Autograft) and IV (BMP) showed unions in the femoral defects.

**Figure 4 bioengineering-10-00086-f004:**
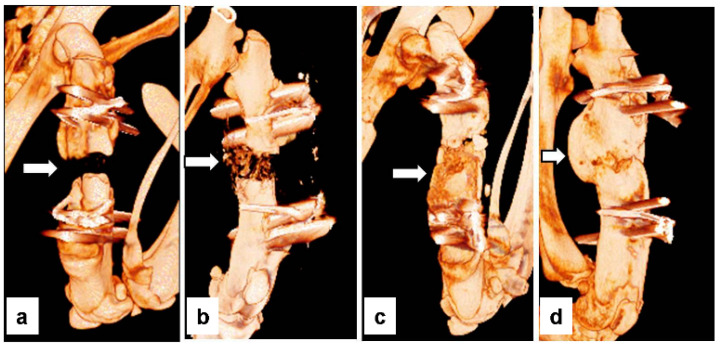
3D reconstruction images showing bone formation at the femoral defect sites in each group. (**a**) group I (Control), (**b**) group II (DFDB graft), (**c**) group III (Autograft), and (**d**) group IV (BMP). No bony tissue was found in the femoral defects in group I, a small amount of bony tissue was found in group II, a large amount of bony tissue was found in group III, and the largest amount of bony tissue was found in group IV.

**Figure 5 bioengineering-10-00086-f005:**
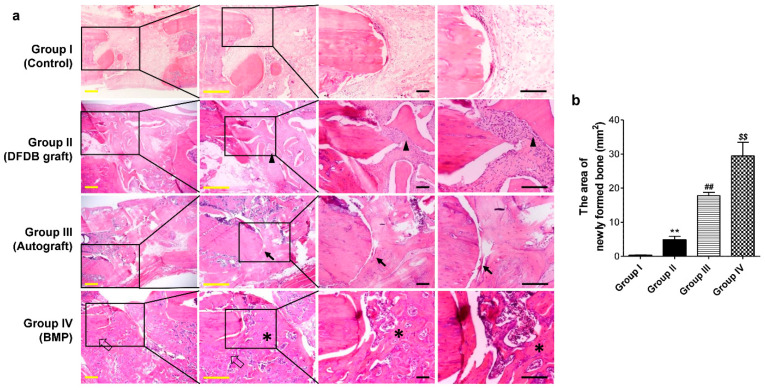
Histologic findings of the bone defect sites in the various experimental groups. (**a**) The bone defect site was filled with fibrous tissue without bone formation in the absorbable collagen sponge alone group. The defect site in the demineralized freeze-dried bone (DFDB) graft group was filled with fragmented lamellar bone (the arrowhead indicates the grafted bone) and fibrous tissue. The defect site in the autograft group was also filled with lamellar bone. However, the lamellar bone in the defect site with an autograft was thicker and more abundant, with relatively less fibrous tissue compared to the DFDB graft group. However, the junction between the cortical bone and newly formed bone from the autograft was partially discontinuous (arrows). In contrast, the defect site in the BMP-treated group was filled with newly formed bone (asterisks), and the junction between the cortical bone and new bone was continuous (empty arrows). Yellow bars = 1 mm, Black bars = 200 μm. (**b**) The measurement of the area of newly formed bone from staining images of tissue sections (n = 10), and the values are means ± SEM. and the values are means ± SEM. ** *p* < 0.01 versus group I, ## *p* < 0.01 versus group II, $$ *p* < 0.01 versus group III.

**Table 1 bioengineering-10-00086-t001:** Radiologic union. Radiologic union was assessed by bony bridge formation of the cortex. A complete union was defined as the bony bridge formation of both cortexes in radiography. A partial union was defined as the bony bridge formation of one cortex. The numbers stated in the table (number/number) represent complete union/partial union. (one-way ANOVA, *p* < 0.05. *Posthoc* test with Scheffe, group I vs. group II, *p* = 0.231; group I vs. group III, *p* < 0.001, group I vs. group IV, *p* < 0.001; group II vs. group III, *p* = 0.035; group II vs. group IV, *p* = 0.044, group III vs. group IV, *p* = 0.766).

Complete/Partial Union	4 Weeks	8 Weeks	12 Weeks	16 Weeks
Group I (Control, n = 10)	0/0	0/1	0/1	0/1
Group II (DFDB graft, n = 10)	0/2	0/2	1/3	2/2
Group III (Autograft, n = 10)	0/3	1/3	4/3	6/3
Group IV (BMP, n = 10)	0/3	1/4	3/4	6/2

**Table 2 bioengineering-10-00086-t002:** Manual test. Manual palpation with varus/valgus and anterior/posterior angulated force was performed for all harvest femora (16 weak after surgical procedure). When no motion was present, it was considered a complete union. (one-way ANOVA, *p* < 0.05. *Posthoc* test with Scheffe, group I vs. group II *p* = 0.072; group I vs. group III, *p* < 0.001; group I vs. group IV, *p* < 0.001; group II vs. group III, *p* = 0.018; group II vs. group IV, *p* = 0.018; group III vs. group IV, *p* = 1.0)

Complete Union	16 Weeks	Union Rate
Group I (Control, n = 10)	0	0
Group II (DFDB graft, n = 10)	2	20%
Group III (Autograft, n = 10)	6	60%
Group IV (BMP, n = 10)	6	60%

**Table 3 bioengineering-10-00086-t003:** Bone structure parameters–Micro CT. Bone morphometric parameters of healing of the defect sites in the femur. Data presented as mean ± SD (one-way ANOVA, *p* < 0.05. Posthoc test with Scheffe, bone volume (BV) group I vs. group II, *p* = 0.304; group I vs. group III, *p* = 0.005; group I vs. group IV, *p* < 0.001; group II vs. group III, *p* = 0.026; group II vs. group IV, *p* < 0.001; group III vs. group IV, *p* = 0.06; bone volume/tissue volume (BV/TV) group I vs. group II, *p* = 0.005; group I vs. group III, *p* < 0.001; group I vs. group IV, *p* = 0.0034; group II vs. group III, *p* < 0.001; group II vs. group IV, *p* = 0.026; group III vs. group IV, *p* = 0.25).

	BV (mm³)	BV/TV (%)
Group I (Control, n = 10)	42.8 ± 19.8	15.79 ± 5.6
Group II (DFDB graft, n = 10)	72.93 ± 15.6	36.44 ± 2.5
Group III (Autograft, n = 10)	168.78 ± 20.1	64.3 ± 1.4
Group IV (BMP, n = 10)	193.94 ± 61.5	56.59 ± 9.8

## Data Availability

Not applicable.

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
