# Peer review of "Bone Morphogenetic Protein 2 Promotes Bone Formation in Bone Defects in Which Bone Remodeling Is Suppressed by Long-Term and High-Dose Zoledronic Acid"

_bioengineering, 2023, doi:10.3390/bioengineering10010086_

Round 1
Reviewer 2 Report
Nothing rilevant to suggest
Author Response
We would like to thank the reviewers for their thorough review of our manuscript.
Reviewer 3 Report
This study focuses on the positive effect of BMP-2 on bone formation in conditions where bone remodeling is inhibited by long-term and high-dose zoledronic acid. This study is highly targeted and the idea is clear, but the data in the manuscript is relatively thin and there are still many problems to be solved. I would like to recommend this manuscript to be published after the following issues are well addressed.
1. In order to show the experimental process more clearly, it is suggested to add a schematic diagram in the manuscript.
2. A separate conclusion is needed at the end of the manuscript.
3. The expression of some words in the manuscript needs to be unified, such as "16 weeks" in Figure 2 and "16 wks" in Table 1 and Table 2.
4. In addition to the staining images of tissue section, the corresponding semi-quantitative analysis (such as the area of newly formed bone) should be included in histological analysis.
5. In histological analysis, since it was necessary to observe the newly formed bone, why did the author treat the tissue sections with hematoxylin staining instead of Goldner’s trichrome staining?
6. Since most humans treated with long-term bisphosphonates are older patients with osteoporosis, why did the author choose 12-week-old rats instead of older rats for this study?
7. Could the author briefly speculate on the possible mechanism of the novel finding (BMP-2 promotes bone formation in bone defects in which bone remodeling is suppressed by long-term and high-dose zoledronic acid) in this study?
8. BMP-2 is also produced in the process of bone healing. Will new BMP-2 be produced in the environment where bone remodeling is inhibited by long-term and high dose of zoledronic acid? Will these newly produced BMP-2 affect the experimental results?
Round 2
Reviewer 3 Report
All the concerns have been well addressed, it can be accepted now.